# Morphological and molecular convergences in mammalian phylogenetics

Zhengting Zou[1] & Jianzhi Zhang[2]

Phylogenetic trees reconstructed from molecular sequences are often considered more reliable than those reconstructed from morphological characters, in part because convergent evolution, which confounds phylogenetic reconstruction, is believed to be rarer for molecular sequences than for morphologies. However, neither the validity of this belief nor its underlying cause is known. Here comparing thousands of characters of each type that have been used for inferring the phylogeny of mammals, we find that on average morphological characters indeed experience much more convergences than amino acid sites, but this disparity is explained by fewer states per character rather than an intrinsically higher susceptibility to convergence for morphologies than sequences. We show by computer simulation and actual data analysis that a simple method for identifying and removing convergence-prone characters improves phylogenetic accuracy, potentially enabling, when necessary, the inclusion of morphologies and hence fossils for reliable tree inference.

[1] Department of Computational Medicine and Bioinformatics, University of Michigan, Ann Arbor, Michigan 48109, USA. [2] Department of Ecology and Evolutionary Biology, University of Michigan, Ann Arbor, Michigan 48109, USA. Correspondence and requests for materials should be addressed to J.Z. (email: jianzhi@umich.edu).

Having a reliable species tree is prerequisite for understanding evolution, which is necessary for making sense of virtually every biological phenomenon. Traditionally, species trees are inferred using morphological, physiological or behavioural characters, collectively called morphological characters hereinafter. The advent of molecular biology supplied numerous molecular characters in the form of DNA and protein sequences, which are often (although not universally) considered more suitable than morphological characters for phylogenetic inference[1–6]. A major reason of this consideration concerns convergence, which refers to repeated origins of the same character state in multiple evolutionary lineages and is a primary source of error in phylogenetic reconstruction. Compared with morphological characters, molecular characters are believed by many (but not all) to be less susceptible to convergence[1,3–5,7–12]. Nevertheless, this belief appears to arise in the early days of molecular systematics when morphological convergence had long been known while molecular convergence had not. Recent genetic and genomic studies, however, revealed a large number of convergences in protein sequence evolution[13–23], casting a doubt on the above belief. Determining whether morphological characters are more prone to convergence than molecular characters is important for several reasons. First, although morphological and molecular trees are often concordant with each other, this is not always the case[2,5,6,24,25]. Knowledge of the relative prevalence of convergence in the two types of characters helps decide which tree is more trustable. Furthermore, it helps decide whether total evidence trees reconstructed jointly from the two types of characters[5,9,10,25–28] are preferred over trees based on any one type. Second, phylogenetic analysis that includes fossils can help understand evolutionary relation, time and process for fossils as well as extant species[2,5,9,10,12,25,26,29]. Because molecular characters are inaccessible in the vast majority of fossils, knowing the frequency of morphological convergence is critical to assessing the reliability of phylogenies involving fossils. Third, convergence is caused by either repeated adaptations of different evolutionary lineages to similar environmental challenges or chance. Recent studies suggested that most molecular convergence events are attributable to chance[18,19,30,31]. A comparison between morphological and molecular characters may provide information about the relative roles of selection and drift in morphological evolution.

Because not all morphological or molecular characters are employed by phylogeneticists, a fair comparison between the two character types in the context of phylogenetics should concentrate on characters used for phylogenetic reconstruction. To this end, we analysed a large data set including 3,414 parsimony informative morphological characters and 5,722 parsimony informative amino acid sites that was previously compiled for the inference of mammalian phylogeny of 46 extant and 40 fossil species[25]. Our analysis focused on extant species because they have both types of characters. We found that morphological characters experience much more convergences than molecular characters. We devised a method to identify and remove convergence-prone characters, enabling the inclusion of morphologies and hence fossils for reliable tree inference.

showed convergence (Fig. 1a; see Methods) and compared the mean number of convergences per character between morphological and molecular characters. For example, under the morphological tree, the exterior branches, respectively, leading to wolf (Canis lupus) and aardvark (Orycteropus afer) form an independent branch pair (Supplementary Fig. 1a), where 0.0072 convergences per morphological character was observed, significantly exceeding that (0.0038) per molecular character ($P = 0.03$, Fisher's exact test; see Methods). Among 3,396 pairs of independent branches in the morphological tree, 79.1% exhibit a higher convergence per morphological character than that per molecular character (Fig. 1b), significantly exceeding the chance expectation ($P < 1 \times 10^{-4}$, bootstrap test; see Methods). There are 645 branch pairs with significantly higher per character morphological convergence than molecular convergence (Q-value $< 0.05$, Fisher's exact test; blue dots in Fig. 1b), whereas the opposite is true for only 61 branch pairs (orange dots in Fig. 1b). The mean number of convergence per morphological character is 1.7 times that per molecular character.

It was proposed that convergence is more fairly compared among characters or branch pairs by the ratio between the number of convergence and that of divergence ($Cv/Dv$; Fig. 1a)[15,30] because both $Cv$ and $Dv$ increase with the amount of evolution. Hence, we identified divergence events for each branch pair (see Methods) and then calculated the total number of convergence events relative to the total number of divergence events for the branch pair for each type of characters. We found that morphological characters exhibit overwhelmingly larger $Cv/Dv$, compared with molecular characters (Fig. 1c). The mean $Cv/Dv$ ratio of morphological characters is 4.0 times that of molecular characters.

If the morphological tree used differs from the unknown true tree, inferring convergence under the morphological tree underestimates morphological convergence and hence the conclusion of a higher convergence for morphological characters than molecular characters should be conservative. As expected, when the above analyses were repeated under the molecular tree (Supplementary Fig. 1b) or the total evidence tree (Supplementary Fig. 1c), we found even higher convergences (Fig. 1d; Supplementary Fig. 2a) and $Cv/Dv$ ratios (Fig. 1e; Supplementary Fig. 2b) for morphological characters than for molecular characters. Similar results were obtained using conventional measures of homoplasy such as the consistency index ($ci$) and rescaled consistency index ($rc$). That is, regardless of the tree topology used, morphological characters show lower $ci$ and $rc$, thus higher homoplasy, than molecular characters (Supplementary Fig. 3).

DNA sequences instead of amino acid sequences are sometimes used as molecular characters in phylogenetics. We, therefore, also conducted a whole-tree analysis of the 19,227 parsimony informative nucleotide sites in the data set, with the tree inferred from the nucleotide sequences as the molecular tree. Regardless of whether the morphological or molecular tree is used, we observed higher convergence per character and higher $Cv/Dv$ ratio for morphological characters than nucleotide sites (Supplementary Fig. 4a–d).

## Results

**Whole-tree analysis**. Analysing character convergence requires a species tree. Because the mammalian tree is not completely resolved, we used three trees, respectively, reconstructed using the morphological characters, molecular characters, and both types of characters in the data set. Under each tree, we inferred the ancestral states at all interior nodes for each character. For every pair of independent branches, we identified characters that

**Quartet analysis**. Because the true mammalian tree is unknown, to ensure a fair comparison between morphological and molecular characters, we further examined every four species in the data that show the same phylogenetic relationship in the morphological and molecular trees, which we refer to as quartets (Fig. 2a). Given a quartet and their phylogenetic relationship, a parsimony-informative character is said to be convergent if at least two changes are required to explain the observed states

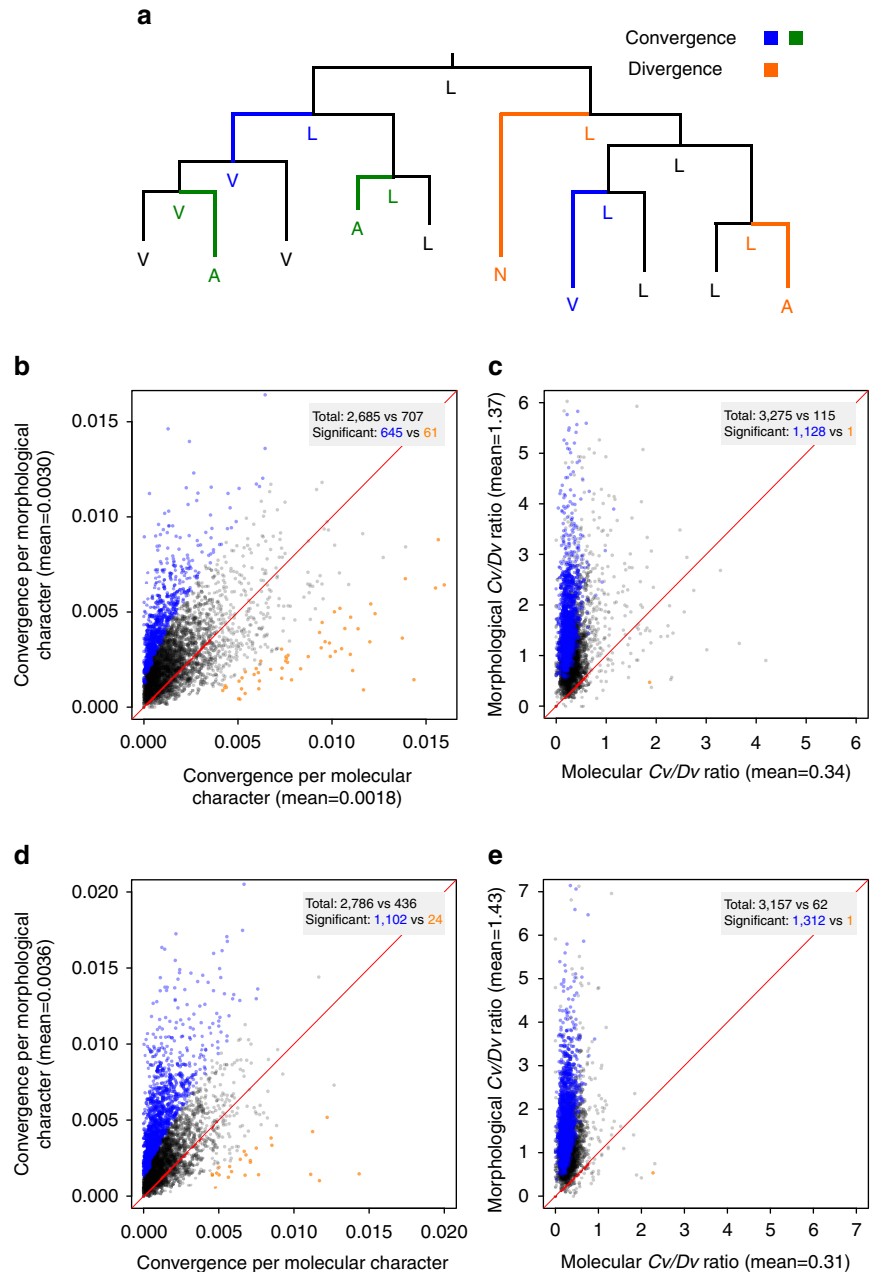

**Figure 1 | Whole-tree analysis of morphological and molecular convergences in mammals.** (**a**) Schematic examples of convergence and divergence. Given the states of the interior and exterior nodes of the tree, the blue and green branch pairs each experienced a convergence event, while the orange branch pair experienced a divergence event. A, L, N and V are four different states of a character. (**b**) Mean number of convergences per morphological character and that per molecular character for each branch pair examined under the morphological tree. (**c**) Convergence/divergence ($Cv/Dv$) ratio for each branch pair under the morphological tree. (**d**) Mean number of convergences per morphological character and that per molecular character for each branch pair examined under the molecular tree. (**e**) $Cv/Dv$ ratio for each branch pair under the molecular tree. In **b**–**e**, each dot represents a branch pair. In the grey box of each panel, 'total' refers to the numbers of dots above and below the diagonal (dots on the diagonal are not counted), respectively, and 'significant' refers to the numbers of dots significantly (at $Q$-value of 0.05) above (blue) and below (orange) the diagonal, respectively. Total number of dots above the diagonal significantly exceeds that below the diagonal in **b**–**e** ($P < 1 \times 10^{-4}$, bootstrap test). For **c** and **e**, branch pairs with infinite $Cv/Dv$ values are not plotted but included in the comparison.

(Fig. 2a). We identified all convergence events for each quartet. Averaged across 7,146 quartets that can be examined, we observed 0.026 convergences per morphological character, which is three times that per molecular character (0.0085). Higher morphological convergence than molecular convergence is found in 93.9% of quartets (Fig. 2b), significantly exceeding the chance expectation ($P < 1 \times 10^{-4}$, bootstrap test). A total of 6,087

quartets show significantly higher per character morphological convergence than molecular convergence ($Q$-value $< 0.05$), while only 104 quartets show the opposite (Fig. 2b).

Given a quartet and their phylogenetic relationship, a parsimony-informative character is said to be consistent when only one change is needed to explain the observed states. Convergence provides an erroneous phylogenetic signal for the

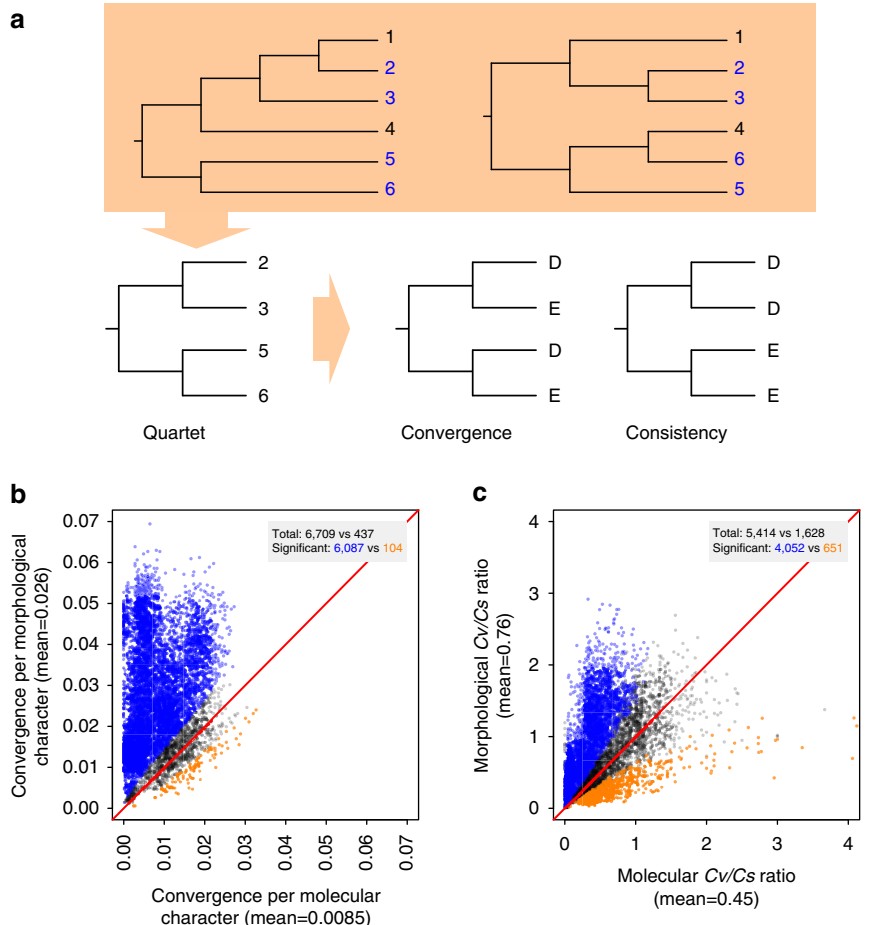

**Figure 2 | Quartet analysis of morphological and molecular convergences in mammals.** (**a**) A schematic example of a quartet, which are four species (2, 3, 5 and 6) showing the same phylogenetic relationship in the morphological (left) and molecular (right) trees. Examples of character states exhibiting convergence and consistency are shown. (**b**) Mean number of convergences per morphological character and that per molecular character for each quartet examined. (**c**) Convergence/consistency (*Cv/Cs*) ratio for each quartet. In **b** and **c**, each dot represents a quartet. In the grey box of each panel, 'total' refers to the numbers of dots above and below the diagonal (dots on the diagonal are not counted), respectively, and 'significant' refers to the numbers of dots significantly (at *Q*-value of 0.05) above (blue) and below (orange) the diagonal, respectively. Total number of dots above the diagonal significantly exceeds that below the diagonal in **b** and **c** ($P<1\times10^{-4}$, bootstrap test). In **c**, quartets with infinite *Cv/Cs* values are not plotted but included in the comparison.

quartet, whereas consistency offers the correct signal. We thus computed, for each quartet, the ratio between the total number of *co*nvergences and that of *co*nsistencies (*Cv/Cs* ratio) for each type of characters, which may be viewed as the noise/signal ratio. Again, morphological characters tend to have higher *Cv/Cs* ratios than molecular characters (Fig. 2c). The above results also hold when nucleotide sites instead of amino acid sites are used as molecular characters (Supplementary Fig. 4e,f).

**Number of states per character.** We found that 75.2% of parsimony-informative morphological characters are binary (Fig. 3a). Because binary characters can only have one kind of change given an ancestral state, it is obvious that they are susceptible to convergence once multiple changes occur. By contrast, only a small fraction (12.4%) of molecular characters are binary (Fig. 3a). The median number of states is five for molecular characters, significantly higher than that (two) for morphological characters ($P<10^{-300}$, Mann–Whitney *U*-test).

The probability of convergence relative to that of divergence for a character is expected to decrease with the number of states. Indeed, the *Cv/Dv* ratio decreases with the number of states for both types of characters (Fig. 3b; Supplementary Fig. 5) and this

trend remains after the control of evolutionary rate (Supplementary Table 1). We estimated that the *Cv/Dv* ratio of an average morphological character is 0.89 times that of a molecular character with the same number of states, and the corresponding number is 0.55 for *Cv/Cs* (see Methods). These results indicate that, compared with molecular characters, the higher convergence of morphological characters is caused by having fewer states rather than intrinsically higher susceptibilities to adaptive convergent evolution, because morphological characters are no more prone to convergence than molecular characters once the number of states is controlled for.

The above patterns remain unchanged even when nucleotide sites instead of amino acid sites are used as molecular characters (Supplementary Table 2). Interestingly, although there can be no more than four states at each nucleotide site, the median number of states (three) per nucleotide site is still significantly higher than that (two) per morphological character ($P<10^{-300}$).

**Removing convergence-prone characters improves phylogenetics.** Because the vast majority of molecular convergences are explainable by chance[18,19,30,31], the fact that average morphological characters have even smaller *Cv/Dv* and *Cv/Cs* ratios

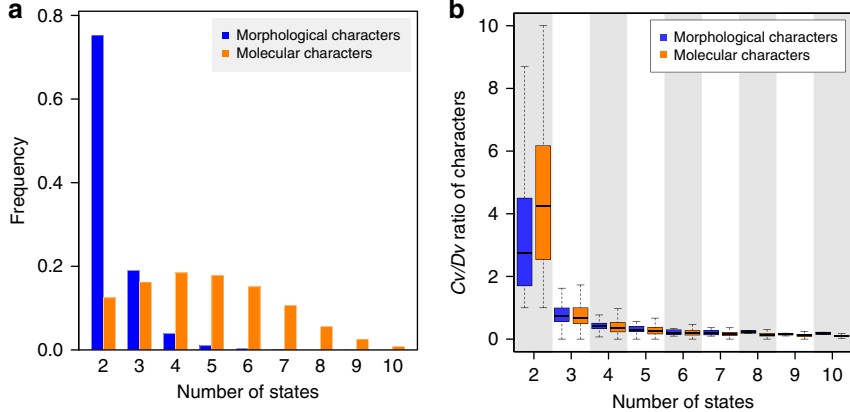

**Figure 3 | Morphological characters tend to have fewer states than molecular characters.** (**a**) Frequency distribution of the number of states per character. (**b**) $Cv/Dv$ ratio of a character decreases as the number of states increases. $Cv/Dv$ ratio of a character is the sum of convergences across all branch pairs divided by that of divergences. The top and bottom edges of a box represent the first and third quartiles of the distribution, respectively, while the thick line inside the box represents the median. The two whiskers show the maximum value not greater than the first quartile plus 1.5 times the box height and the minimum value not smaller than the third quartile minus 1.5 times the box height, respectively. $Cv/Dv$ ratios are calculated under the morphological tree. The same pattern is observed when $Cv/Dv$ ratios are calculated under the molecular tree (Supplementary Fig. 5).

than those of molecular characters of the same numbers of states suggest that most morphological convergences observed in the data analysed are probably also attributable to chance. If convergence is owing to chance rather than lineage-specific selection, it is possible to identify and remove convergence-prone characters using species with reliable phylogenetic relations and then infer the tree for species of uncertain relations using the remaining characters. This approach would be especially beneficial to phylogenetic inference that includes morphological data because of the relatively frequent convergence in such data. We propose the following procedure when analysing a data set with both morphological and molecular characters. First, we infer the morphological and molecular trees separately. Second, quartets (that is, groups of four species with the same phylogenetic relations in the two trees) are identified and the $Cv/Cs$ ratio is calculated based on these quartets for each character. Third, we remove all characters whose $Cv/Cs$ ratio exceeds a cutoff and infer the tree using all remaining morphological and molecular characters combined.

To investigate whether the above approach improves phylogenetic accuracy, we conducted 50 simulations of mammalian morphological and molecular characters based on their respective empirical distributions of the number of states (Supplementary Fig. 6a). Quartet analysis demonstrates that the simulated data have similar properties as the real data (Supplementary Fig. 6b,c; Supplementary Table 3). We measured the Robinson-Foulds distance ($d_{RF}$) between an inferred tree and the known true tree in simulation; $d_{RF}$ is twice the fraction of branch partitions that differ between the two trees[32]; the smaller the $d_{RF}$, the more accurate the inferred tree. We found that $d_{RF}$ is significantly greater for the 50 morphological trees than the 50 molecular trees ($P = 1.6 \times 10^{-14}$, Mann–Whitney $U$-test), confirming the damage of random convergence on phylogenetic accuracy. We set 10 $Cv/Cs$ cutoffs from 5 to 0.03 and inferred ten low-convergence total evidence trees for each simulated data set using the above proposed procedure (see Methods). We found that $d_{RF}$ to the true tree is generally smaller for low-convergence trees than the original tree reconstructed using all characters (green symbols in Fig. 4a), and the improvement in phylogenetic accuracy plateaus when the cutoff reaches 0.3. By contrast, trees based on a

random removal of the same number of characters do not show smaller $d_{RF}$ when compared with the original tree (pink symbols in Fig. 4a). As expected, the mean number of states is higher for the remaining low-convergence characters than for the same number of characters randomly picked from the original simulated data (Supplementary Fig. 6d).

**Removing convergence-prone characters alters the bat tree.** We applied the above pipeline to the mammalian data set including both morphological characters and amino acid sequences. The same 10 $Cv/Cs$ ratio cutoffs as in the simulation were used in removing high-convergence characters, and low-convergence total evidence trees of all 86 extant and fossil species were inferred using the remaining morphological and molecular characters. For the 46 extant species that can be compared, the resultant low-convergence trees are generally more similar than trees based on the same numbers of randomly selected characters to the original molecular tree (Supplementary Fig. 7). The low-convergence trees are also generally more different than trees based on the same numbers of randomly selected characters from the original morphological tree (Supplementary Fig. 7). Although the true mammalian tree is unknown, these observations are consistent with our finding that convergence is less frequent in molecular characters than morphological characters.

Regarding intra-order relationships, the phylogeny of bats has been highly controversial. Specifically, all echolocating bats typically form a monophyletic group in morphological trees, suggesting a single origin of bat echolocation[25]. But they tend to form a paraphyly in molecular trees[33–36], suggesting the possibility of two origins of bat echolocation or one origin followed by a loss. In the original total evidence tree (Supplementary Fig. 8a) reconstructed using the data analysed here, all five extant species of echolocating bats form a monophyly to the exclusion of the only non-echolocating extant bat *Pteropus giganteus*, with a 99.2% bootstrap support (Fig. 4b). When the 3,930 characters (1,007 morphological and 2,923 molecular) with $Cv/Cs$ ratio $<0.2$ are used after the removal of 2,407 morphological characters and 2,799 molecular characters (Supplementary Fig. 8b), echolocating bats become paraphyletic; the echolocating *Rhinopoma hardwickii* and non-

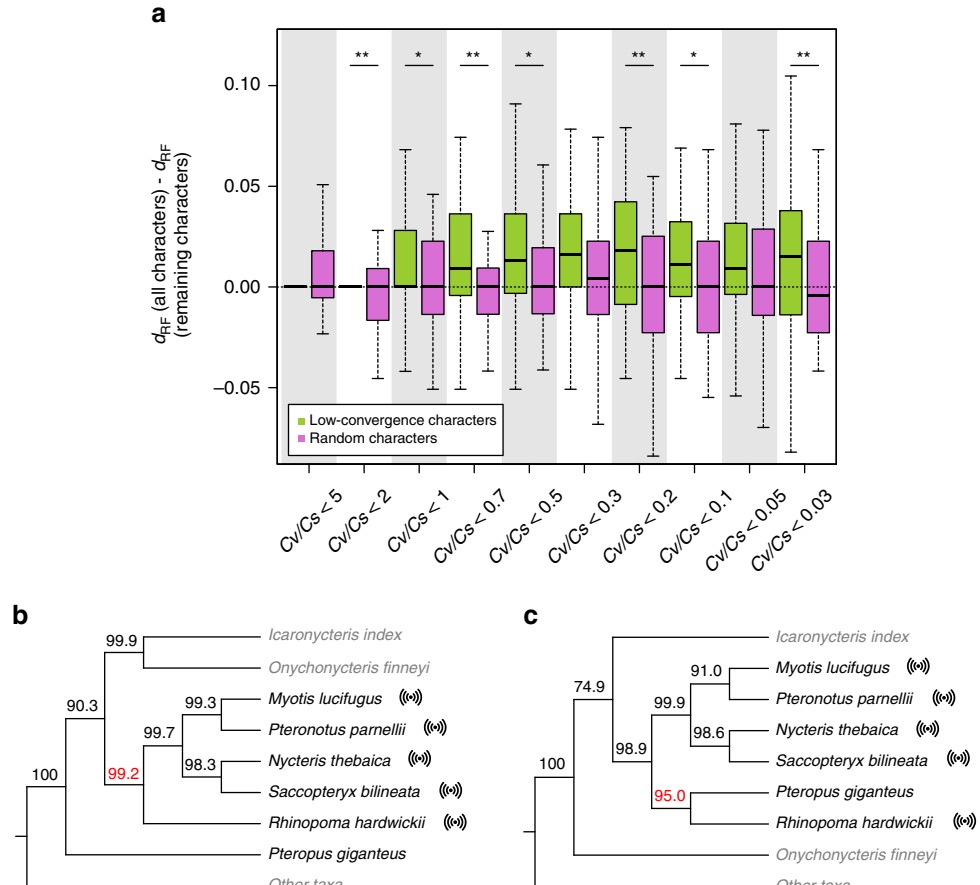

**Figure 4 | Removing convergence-prone characters improves phylogenetic accuracy.** (**a**) Simulation results showing that using characters with $Cv/Cs$ ratios below certain cutoffs reduces the Robinson-Foulds distance ($d_{RF}$) between the true tree and the inferred tree, while using the same number of randomly picked characters does not. The top and bottom edges of a box, respectively, represent the first and third quartiles of the distribution from 50 simulations, while the thick line inside the box represents the median. The two whiskers show the maximum value not greater than the first quartile plus 1.5 times the box height and the minimum value not smaller than the third quartile minus 1.5 times the box height, respectively. $Cv/Cs$ ratios are estimated based on quartets (sets of four species with the same phylogenetic relationships in the inferred morphological and molecular trees of the simulated data). *$P < 0.05$, paired Mann–Whitney $U$-test from 50 simulations; **$P < 0.01$. (**b**) Bat relationships based on the parsimony tree reconstructed using 9,136 informative morphological and molecular characters. Echolocating species are marked with an echo sign. Extant bats are in black, while fossil bats and other taxa are in grey. Bootstrap percentage is presented for each internal node, with the red colour highlighting the bootstrap percentage for the monophyly of echolocators. (**c**) Topology based on 3,930 informative characters with $Cv/Cs$ ratios $< 0.2$. The red colour highlights the bootstrap percentage for the sister relationship between the non-echolocator *Pteropus giganteus* and the echolocator *Rhinopoma hardwickii*.

echolocating *P. giganteus* are grouped with a 95.0% bootstrap support (Fig. 4c). Note that using low-convergence morphological characters alone does not result in this new topology. For comparison, we generated 50 randomly subsampled data sets, each with 1,007 morphological and 2,923 molecular characters. Although 18 of them also yielded the same topology as in Fig. 4c, the corresponding bootstrap support ranged between 18 and 70%, suggesting that the strong support for the paraphyly of echolocators in Fig. 4c is not explained simply by subsampling of the original data. Our results are not sensitive to the $Cv/Cs$ ratio cutoff, because the same bat relationships were recovered when any $Cv/Cs$ cutoff of 0.3 or smaller was used.

## Discussion

Our analysis of comparably large numbers of morphological and molecular characters previously used in inferring the mammalian tree showed that morphological characters experienced more convergent evolution than molecular characters, confirming a long-held belief of the phylogenetics community. Nevertheless,

we caution that our conclusion should be further scrutinized using additional data from additional groups of species, because they are currently based on only one, albeit very large, data set of one group of species. There are three potential sources of error in our inference of convergence. First, use of a wrong species tree could bias our inference. But, as demonstrated, our results are robust to different species trees used. Second, our inference of convergence relies on ancestral state reconstruction by parsimony that may contain errors[37]. But, such errors should be comparable between the two types of characters. Third, it was recently proposed that some inferred convergences may be caused by incomplete lineage sorting rather than genuine convergent changes[38]. Similar to genuine convergence, apparent convergence owing to incomplete lineage sorting also confounds phylogenetic inference and thus need not be separated from our estimates of convergence. Hence, the three potential errors do not affect our conclusion.

Regarding the reason behind the higher convergence of morphological characters than molecular characters, our results do not support the common view that morphological characters

are intrinsically more prone to convergence because they are more frequently subject to positive selection. Instead, we found the probability of convergence for a character to decrease with the number of states and found no greater intrinsic propensities for convergence (as measured by $Cv/Dv$ and $Cv/Cs$ ratios) among morphological characters than molecular characters after the control of the number of states. A likely explanation for this unexpected finding is that phylogeneticists have removed morphological characters that are subject to frequent positive selection (for example, body size and coat colour) from phylogenetic analysis, because such characters are known to lack reliable phylogenetic signals[39]. As a result, the morphological characters used for phylogenetic inference have relatively low intrinsic propensities for convergence. If most convergences of the morphological characters in the data analysed are not manifestations of repeated adaptations but pure chance, one wonders what morphological characters are responsible for the clustering of species with seemingly adaptive convergences in the morphological tree, such as the clade of the four ant- and termite-eaters: the nine-banded armadillo *Dasypus novemcinctus*, collared anteater *Tamandua tetradactyla*, Chinese pangolin *Manis pentadactyla*, and aardvark *Orycteropus afer* (Supplementary Fig. 1a). These species form three independent lineages (*Dasypus + Tamandua*, *Manis*, and *Orycteropus*) in the molecular tree (Supplementary Fig. 1b) as well as the total evidence tree (Supplementary Fig. 1c). We found that, even on the basis of the molecular tree, at most 14 morphological characters are inferred to have experienced convergence among the three lineages, and the actual number is likely much smaller because, for 13 of the 14 characters, convergence is but one of several equally parsimonious evolutionary scenarios. However, none of the 14 characters are apparently related to ant- and termite-eating or are specific to these four species. For instance, the only character for which the sole parsimonious reconstruction indicates convergence among the three lineages describes the shape of the medial border of humerus trochlea. The humerus is a long bone in the arm or forelimb that runs from the shoulder to the elbow and trochlea refers to a grooved structure reminiscent of a pulley's wheel. This character does not appear to be related to ant- and termite-eating. In fact, manatee (*Trichechus manatus*) and ring-tailed lemur (*Lemur catta*) also have the same state as the four ant- and termite-eating mammals for this character. These findings are consistent with our conclusion that most morphological convergences observed here are caused by chance rather than repeated adaptations. Of course, we cannot exclude the possibility that a small number of morphological convergences observed in this data set are adaptive.

Nevertheless, morphological characters experience more convergences than molecular characters, because of much fewer states in the former than the latter. The low number of states per morphological character may be related to one or both of the following reasons[7,10]. First, curating multistate morphological characters may be more subjective and error-prone, resulting in a reduced use of such characters in phylogenetics[40]. Second, most morphological characters may have a small state space, rendering finding multistate characters difficult[41].

Because of the higher prevalence of convergence among morphological characters than molecular characters and the rapid accumulation of molecular sequence data, we suggest that phylogenetic reconstruction should normally use only molecular data. In the event that molecular data are inaccessible for some taxa such as fossils, one should consider using morphological characters with relatively large numbers of states to minimize convergence in phylogenetic analysis.

Given a data set of morphological and molecular characters, we proposed a method to reconstruct more accurate total evidence trees by identifying and removing convergence-prone characters in the data set, and demonstrated its validity by computer simulation. Homoplasy, which interferes with phylogenetic inference, also includes reversal in addition to convergence. While our study focuses on convergence, it is worth noting that convergence-prone characters are also expected to be reversal-prone if most convergences are chance events owing to the availability of only few states, as indicated by the present data. Thus, in removing convergence-prone characters, we effectively also take out many reversal-prone characters; the success of our method may be in part attributable to this effect. Because our method relies on the assumption that characters that are convergence-prone in the quartets analysed are also convergence-prone in other species, it is not effective in removing characters that are convergence-prone in a few specific lineages such as those subject to adaptive convergence. In principle, one could also downweight instead of removing convergence-prone characters, but the appropriate weights are unknown. Future studies can investigate how to acquire the best weights for improving phylogenetic accuracy.

We showed that the original total evidence mammalian tree in which all echolocating bats form a monophyly is altered upon the removal of convergence-prone characters. The low-convergence tree shows a paraphyly of echolocating bats, identical to the recently published genome-based bat phylogeny[34]. Assuming that the genome-based tree is correct, our results demonstrated the utility of our method in actual phylogenetic inference with the total evidence approach. Besides, our low-convergence tree also supports the monophyly of pangolin (*Manis pentadactyla*) and carnivores (Supplementary Fig. 8b), which is not reflected in the original total evidence tree (Supplementary Fig. 8a) but is supported by previous molecular studies[33,42]. As shown by our computer simulation, although removing convergence-prone characters improves phylogenetic accuracy, low-convergence trees may still contain errors. Identifying and removing convergence-prone characters is by no means a panacea for phylogenetics. While rapidly accumulating genome sequences will eventually dwarf the morphological data of any extant species, morphological data will remain useful in phylogenetic analysis that needs to contain fossils, whose value to understanding evolution is indispensable. For this reason, understanding and remedying convergence, which is more prevalent in morphological than molecular characters, will remain an important task in phylogenetics. Of course, morphological characters that can be studied in fossils do not represent a random sample of all morphological characters. Whether this nonrandomness will bias phylogenetic inference[43] is also worth investigation.

## Methods

**Data set used.** The original data set is composed of 4,541 morphological characters and 11,365 amino acid sites[25]. It includes 86 species, with 40 fossil taxa having only morphological characters and 46 extant species having both types of characters. We focused on extant species in this study because they have both types of characters for comparison. The morphological tree, molecular tree, and total evidence tree (that is, based on both types of characters) built using the parsimony method were provided by the original study (see Supplementary Fig. 1). We removed all parsimony-uninformative characters for the 46 extant species. A parsimony-informative character has at least two states, each represented by at least two taxa. Parsimony trees of the 46 extant species based on the remaining 3,414 morphological characters or 5,722 amino acid sites agree with those based on all characters of the same types.

**Whole-tree analysis.** Ancestral states of each parsimony-informative character were inferred for all interior nodes in the morphological tree, molecular tree or total evidence tree by parsimony using Mesquite (V. 3.03) (http://mesquiteproject.org/). Equal weights were given to equally parsimonious pathways in counting convergence events, such that if only one of $n$ equally parsimonious pathways for a character shows convergence for a branch pair, $1/n$ convergence events are counted. Missing extant states of a character were inferred simultaneously during

the inference by parsimony, such that no additional changes are required due to the missing state assignment. Mesquite also output the number of states appearing in the 46 extant species for each character and the number of changes each character experienced along the entire tree.

An independent branch pair refers to two branches that are not ancestral to each other and contain no common node. For example, let the starting and end states of one branch (node 1 to node 3) be $X_1$ and $X_3$, and let those of another branch (node 2 to node 4) be $X_2$ and $X_4$, respectively. These two branches form an independent branch pair if (i) the four nodes are all distinct from one another, (ii) node 3 is not on the path from the tree root to node 4 and (iii) node 4 is not on the path from the tree root to node 3. For an independent pair of branches, there is a convergence if and only if $X_1 \neq X_3$, $X_2 \neq X_4$, and $X_3 = X_4$. This definition includes both parallel and convergent changes previously defined[13]. Similarly, there is a divergence in the independent branch pair if and only if $X_1 \neq X_3$, $X_2 \neq X_4$, and $X_3 \neq X_4$. Thus, once ancestral states are inferred, we know whether a character experiences convergence, divergence, or neither for a branch pair. For a character, the consistency index ($ci$) is the smallest minimal number of changes required to explain the observed states by any tree (Min) divided by the minimal number of changes required by the tree under evaluation (Obs). Retention index ($ri$) = (Max − Obs)/(Max − Min), where Max is the largest minimal number of changes required to explain the observed states by any tree. Rescaled consistency index ($rc$) equals consistency index multiplied by retention index[44]. Values of $ri$ and $ci$ were calculated by Mesquite.

We used Fisher's exact test to compare the number of convergences per character or $Cv/Dv$ ratio between morphological and molecular characters. For example, in the branch pair leading to wolf and aardvark, we inferred 24.67 convergences among 3,414 morphological characters and 21.88 convergences among 5,722 molecular characters. We rounded the decimal number of convergence to the nearest integer and tested the null hypothesis that the probability of experiencing convergence is the same for the two types of characters using the following 2 × 2 contingency table: 25, 3414-25, 22 and 5722-22. We obtained a two-tailed $P$ value of 0.0332 using Fisher's exact test, indicating that the frequency of convergence is significantly higher for morphologies than for sequences for the branch pair. For the same branch pair, we inferred 9.84 and 88.57 divergence events for morphological and molecular characters, respectively. We tested the null hypothesis that $Cv/Dv$ ratio is the same for the two types of characters using the contingency table of 25, 10, 22 and 89. The obtained two-tailed $P$ value from Fisher's exact test is $3.9 \times 10^{-8}$, indicating that $Cv/Dv$ ratio is significantly higher for morphological characters than for molecular characters for this branch pair. There were two branch pairs with no convergence and no divergence for molecular characters under the morphological tree, and three such branch pairs under either the molecular tree or the total evidence tree. These branch pairs had undefined $Cv/Dv$ ratios for molecular characters and could not be tested in Fisher's exact test. Hence, they were excluded from the analysis and corresponding figures.

Because branch pairs (or quartets) are not independent from one another, simple parametric statistic tests cannot be used. We thus used a bootstrap method to test the null hypothesis that per character convergence is lower for morphological characters than for molecular characters. First, we generated one bootstrap sample containing the same number of both morphological and molecular characters as in the original data. Second, we analysed all branch pairs using the bootstrap sample and examined if >50% branch pairs show a lower morphological convergence than molecular convergence. We repeated the above two steps 10,000 times and computed the fraction of bootstrap samples in which >50% branch pairs show a lower morphological convergence than molecular convergence. This fraction is an estimate of the probability that the null hypothesis is correct, hence is the $P$ value of this bootstrap test. The same bootstrap method was used to test the null hypothesis that $Cv/Dv$ ratio and $Cv/Cs$ ratio is lower for morphological characters than molecular characters in respective analyses.

**Quartet analysis.** Four extant taxa $Y_1$, $Y_2$, $Y_3$ and $Y_4$ are selected if they satisfy the following conditions: (i) $Y_1$ and $Y_2$ form a monophyletic group in exclusion of $Y_3$ and $Y_4$ in both the morphological and molecular trees of all extant taxa examined; (ii) $Y_3$ and $Y_4$ form a monophyletic group in exclusion of $Y_1$ and $Y_2$ in both the morphological and molecular trees; and (iii) the root of this four-species tree is located on the internal branch in both the morphological and molecular trees. Mapping a parsimony-informative character onto this quartet tree, we say that the character shows a convergence if the states of ($Y_1$, $Y_2$, $Y_3$, $Y_4$) are (A, B, A, B) or (A, B, B, A), where A and B are two observed states of the character in the four species. By contrast, we say that the character shows a consistency if (A, A, B, B) is observed. Statistical tests followed those in the whole-tree analysis, except that quartets replaced branch pairs. There were 103 quartets with zero convergence and zero consistency for molecular characters. These quartets had undefined $Cv/Cs$ ratios for molecular characters and could not be tested by Fisher's exact test. Hence, they were excluded from the analysis and corresponding figures.

**Comparison of $Cv/Dv$ given the number of states.** The $Cv/Dv$ ratio of a character is calculated by the sum of $Cv$ across all branch pairs divided by the sum of $Dv$ across all branch pairs for the character. Morphological and molecular characters are divided into bins according to the number of states. For each bin, a ratio

between mean morphological $Cv/Dv$ and mean molecular $Cv/Dv$ is calculated. Finally, this ratio is averaged across bins, weighted by the number of morphological characters in each bin. Hence, the weighted average reflects $Cv/Dv$ of morphological characters relative to that of molecular characters of the same numbers of states. When the measure of $Cv/Cs$ is used, the same procedure is followed except that quartets instead of branch pairs are used.

**Simulation of character evolution.** The evolution of morphological and molecular characters was simulated according to Markov processes, based on the tree topology and branch lengths of the nucleotide maximum likelihood tree from the original study[25] (Supplementary Fig. 6a). The Newick format of the tree is ((((((((1:0.0759,(2:0.0568,3:0.0467)47:0.0234)48:0.00318,(((4:0.0448,5:0.0626)49:0.00468,((6:0.0656,7:0.0707)50:0.00570,(8:0.0634,9:0.0616)51:0.00210)52:0.0142)53:0.0150,((10:0.0602,(((11:0.0383,(12:0.0233,13:0.0128)54:0.0165)55:0.00491,14:0.0721)56:0.00919,15:0.0666)57:0.00518)58:0.0222,16:0.0559)59:0.00143)60:0.000543)61:0.00249,(17:0.1007,(18:0.0849,(19:0.1390,20:0.1468)62:0.0152)63:0.00163)64:0.00940)65:0.0110,((21:0.0989,22:0.0676)66:0.0303,(23:0.1102,((24:0.0777,25:0.1875)67:0.00944,(26:0.0941,27:0.1660)68:0.00225)69:0.0131)70:0.00528)71:0.00149,((28:0.0618,(29:0.0913,(30:0.0414,31:0.0231)72:0.0368)73:0.00438)74:0.00775,(32:0.00806,33:0.00966)75:0.0581)76:0.00177)77:0.00997)78:0.0107,(34:0.0664,35:0.0869)79:0.0309)80:0.00230,((36:0.0429,(37:0.1000,38:0.0439)81:0.00304)82:0.0133,(39:0.0695,((40:0.1506,41:0.0760)83:0.00679,42:0.1275)84:0.00331)85:0.00283)86:0.0300)87:0.1278,(43:0.0834,44:0.0739)88:0.1754)89:0.1518,(45:0.0454,46:0.0378)90:0.1518)91:0.0000. In the simulated evolution, a model equivalent to the Jukes–Cantor model assuming equal equilibrium frequencies of all states and equal exchange rates among all states was used. For each morphological character, its number of states $N$ is a randomly drawn number from the empirical distribution of the number of states in the original morphological data (Fig. 3a). The 1 PAM transition matrix for this character is an $N \times N$ square matrix $M$ with each non-diagonal item equal to 0.01/$N$. The relative evolutionary rate $r$ of the character is randomly drawn according to a Pearson correlation of 0.64 with the number of states $n$, as was observed in the actual data. Specifically, we draw a random variable $n'$ from the empirical distribution of the number of states and compute $r = 0.64n + n'\sqrt{1 - 0.64^2}$. We then normalize $r$ such that the mean $r$ from all characters equals 1.

The character evolution then starts from a random initial state at the tree root and evolves by a Markov chain along tree branches. Molecular characters were similarly simulated. Fifty simulations were conducted, each composed of 20,000 morphological characters and 40,000 molecular characters. The number of states used to generate each character and the number of substitution steps in evolution were recorded for downstream analysis. Quartet analysis based on a randomly picked simulation showed that the properties of these characters resemble those of the real data. Specifically, for almost all quartets, convergence per character and $Cv/Cs$ ratio are higher for morphological characters than molecular character (Supplementary Fig. 6b,c). In addition, a significant negative partial correlation was observed between the number of states and $Cv/Cs$ ratio when the number of steps was controlled (Supplementary Table 3).

**Inference of parsimony trees.** Because the evolutionary models of morphological characters have not been well established, model-based tree inference is not used here. Instead, we inferred maximum parsimony trees using PAUP4.0 (http://people.sc.fsu.edu/∼dswofford/paup_test/) for both morphological and molecular data to allow fair comparisons. When analysing the real data, 1,000 replicated heuristic searches were performed with parameters from the original study[25]. All fossil taxa were included when morphological characters were used in the inference. Consensus trees were derived when multiple equally parsimonious topologies were found, with a strict collapse of branches and equal weights of all topologies. In the analysis of simulated characters, 5,000 replications were used instead of 1,000. Bootstrap tests were conducted in PAUP with 1,000 replicates unless otherwise mentioned. Bootstrap values were calculated and mapped by custom Python scripts; equal weights were given to all equally parsimonious trees resulting from each bootstrapped data set.

**Phylogenetic analyses of low-convergence characters.** We used various $Cv/Cs$ ratio cutoffs to remove characters whose $Cv/Cs$ ratios are higher than the cutoffs. For example, in the real data of 9136 parsimony-informative characters, 5,206 characters showed $Cv/Cs > 0.2$, according to quartet analysis. Hence, under the cutoff of $Cv/Cs = 0.2$, we retained 9,136 − 5,206 = 3930 characters for tree inference, including 1,007 morphological and 2,923 molecular characters. As a control, we randomly drew 1,007 morphological and 2,923 molecular characters from all 9,136 characters and conducted a phylogenetic analysis. This control was repeated 50 times.

**Data availability.** All morphological and molecular data analysed were previously published[25], and the data matrices and related files were retrieved from MorphoBank Project 773 (http://www.morphobank.org/).

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

## Acknowledgements

We thank Wei-Chin Ho, Bryan Moyers and Jian-Rong Yang for constructive comments. This work was supported in part by a research grant from the U.S. National Institutes of Health (GM103232) to J.Z.

## Author contributions

J.Z. conceived the project; Z.Z. and J.Z. designed the project; Z.Z. conducted the experiments and analysed the data; Z.Z. and J.Z. wrote the manuscript.

## Additional information

**Competing financial interests:** The authors declare no competing financial interests.

