## [Peer Review File · Nature Communications]

Reviewers' comments:

Reviewer #1 (Remarks to the Author):

This is a well written manuscript that outlines a new approach for improving the accuracy of phylogenetic analyses with morphological data. While the general approach is interesting, the analyses are insufficient to justify the main conclusions. I have several comments/suggestions for improvement. Some are major and other are minor.

Major points

1. In the first sentence of the Discussion the authors state the following: "Our analysis of comparably large numbers of morphological and molecular characters previously used in phylogenetic inference showed that morphological characters experienced more convergent evolution than molecular characters, confirming a long-held belief of the phylogenetics community." This conclusion is not general and is based on the authors' comparison of morphological data and amino acid characters from O'Leary et al.'s (2013) mammalian data set. However, not all types of molecular data are the same and it is not clear that this conclusion holds for different types of molecular data. The authors should perform a comparison for morphological and DNA data from O'Leary et al. to determine if the same pattern holds. Unlike amino acids, which have 20 possible character states, there are only four possible character states (nucleotides) for DNA and one may expect to find higher levels of convergence than for amino acids. Also, the majority of substitutions at the DNA level for protein-coding genes are synonymous (silent) substitutions that do not result in a different amino acid. Together, a smaller number of possible character states (4) plus silent substitutions may result in higher levels of convergence for DNA characters than for morphological characters. This should especially be the case for third positions of DNA codons since these will have the highest levels of silent substitutions. The authors should perform a separate analysis with 3rd codon positions. If the tree based on 3rd positions remains closer to the full molecular tree (based on 1st, 2nd, and 3rd positions) but has similar or higher levels of convergence than morphological characters then the authors will need to re-evaluate their conclusions.

2. After proposing a new method to filter out morphological characters with high levels of homoplasy, the authors apply this method to O'Leary et al.'s morphological data set and conclude that it is useful for improving the performance of phylogenetic analyses with morphological characters. However, they only show the tree for bats. The authors should show the full tree based on their "curated" morphological data set. Does their method eliminate major problems with O'Leary et al.'s morphological tree that were discussed by Springer et al. (2013, reference 1 in Zou and Zhang manuscript)? These problems include numerous polyphyletic groups such as an "insectivore" group, a "spiny hedgehog" group, an "ant & termite eating" group, a "tree dwelling group", "an ungulate group", and a "cud-chewing" group. Further, does the "curated" morphological data set result in the recover of natural clades such as Afrotheria, Boreoeutheria, Euarchontoglires, and Laurasiatheria? It is important to show the full results so that the reader can determine if the approach proposed by the authors is generally effective for improving cladistic analyses with morphological characters. I would further suggest that the authors apply their method to another very large morphological data set for mammals to test its general

effectiveness for improving the accuracy of cladistic analyses with morphological data. Specifically, the discrete morphological data set of Halliday et al. (2015, Biol. Rev. doi: 10.1111/brv.12242). Halliday et al.'s tree based on morphology alone has numerous problems (e.g., polyphyly of Afrotheria, Laurasiatheria, and Euarchontoglires; talpid diphyly; elephant shrew diphyly, etc.) and it would be useful to know if the authors' new approach is useful for fixing these problems.

3. Page, 7. "Because the vast majority of molecular convergences are explainable by chance, the smaller Cv/Dv and Cv/Cs ratios of average morphological characters when compared with molecular characters of the same numbers of states suggests that most morphological convergences observed in the data analyzed are probably also attributable to chance." The logic behind this statement is not transparent. Instead, the major problem with morphological data sets such as O'Leary et al. may have more to do with the non-random pattern of homoplasy rather than the amount of homoplasy. Specifically, is homoplasy randomly distributed across the tree or are there correlated patterns of homoplasy that drive the overall results? The recovery of polyphyletic groups such as the "ant & termite eating" group (and others, see above) suggests that correlated homoplasy (i.e., convergence not by chance) is perhaps the major problem with morphological data. For example, ant and termite eating forms tend to have reduced or absent teeth, long snouts, long tongues, forelimbs adapted for digging, etc. It is not clear how Cv/Dv or Cv/Cs indices can be used to differentiate between the amount of homoplasy and the pattern (random versus non-random) of homoplasy.

4. My understanding of the simulations performed by the authors is that they were based on random homoplasy and ignored potential problems with correlated homoplasy as may occur for real morphological data.

Minor point

5. The authors should comment on their results in the context of Sansom and Wills' 2013 paper (Sci. Reports 3, 2545) on stem-ward slippage, where fundamental taphonomic biases for skeletal characters can cause fossils to be recovered as erroneously primitive.

Reviewer #2 (Remarks to the Author):

This is a good paper containing important results that will be of great and widespread interest. I hope that it will find a home in Nature Communications. There are several issues that I think the authors should consider, however.

A Summary of the key results

The authors demonstrate that their sample of morphological characters contain more homoplasy on average than molecular ones. They also show that this is a function of a more limited character space in the case of the former. This last is a surprising result, as it has often been assumed that signal saturation and multiple hits are problems more particularly bedeviling nucleotide data than morphological character states (where the possibilities for character state

innovation are usually believed to be more open-ended). In that context, I'm not sure about the statement in the abstract that convergence "...is widely believed to be rarer for molecular sequences than for morphologies."

B Originality and interest: if not novel, please give references

This is an original contribution, and will be of great interest to all evolutionary biologists and systematists in particular. It has often been assumed that molecular characters have more desirable properties than morphological characters for phylogenetic inference, not least their abundance, some elements of superior objectivity and increasingly sophisticated frameworks for their analysis. This paper demonstrates that this does indeed seem to be the case, and represents an important result entirely appropriate for publication in Nature Communications.

C Data & methodology: validity of approach, quality of data, quality of presentation

The main empirical conclusions of the paper are derived from a single, albeit very large, data set for mammals. The simulations bolster these conclusions, but the authors may wish to consider whether there might be benefits from analyzing a wider sample of empirical cases. I am not suggesting that they should do this, but I think there needs to be some discussion of this limitation in the paper.

D Appropriate use of statistics and treatment of uncertainties.

Subject to C above, the stats and methods seem sound.

E Conclusions: robustness, validity, reliability

Again, one might wish to add some circumspect qualification that this is just a single empirical data set. It is entirely possible - even probable - that the pattern they observe is a general one. I wonder whether their conclusions (and the title of the paper) might be better couched in terms of mammalian phylogeny, for which they have the largest and most impressive data set available. They could then go on to set out the case that this likely to be a generality. Investigating this possibility really is a hugely pressing research agenda, and this paper makes a fantastic, novel and excellent contribution to it.

F Suggested improvements: experiments, data for possible revision

Analysing more data sets is probably impractical, and represents a lot of additional work that I think it would be unfair to ask the authors to do.

Perhaps there needs to be some exploration of how their approach for removing convergence-prone characters compares with methods that remove or otherwise downweight characters prone to homoplasy more generally. Implied weighting, successive approximations weighting and even clique analysis have this effect with varying degrees of severity, in addition to the normal practice of omitting morphological characters a priori that are believed to largely convey noise at the level of the analysis. How do these approaches compare? I am not necessarily suggesting any additional analyses here. Homoplasy comprises convergence plus reversal, and there may be a nuanced argument that the former is intrinsically less likely than the other: hence the advantage of removing convergence-prone characters rather than those subject to high levels of homoplasy more generally. 'Dollo' parsimony, for example, forces all homoplasy to take the form of reversals, and requires a rooted tree rather than a network for its computation.

G References: appropriate credit to previous work?
Yes, all fine here.

H Clarity and context: lucidity of abstract/summary, appropriateness of abstract, introduction and conclusions

Good. I had to read the paper right through before I understood the approach and what had been done properly (with some issues outstanding) and it may be possible to make this clearer earlier in the manuscript.

In the introduction, the authors could state more clearly whether or not they are in favour of using morphological characters. Convergence isn't just a problem for morphological data. It's one of the main issues bedeviling all phylogenetic analyses. The authors also make the point that most morphological and molecular trees are concordant, suggesting the situation is less black and white than some of their other statements might suggest.

The method for assessing the amount of convergences on branches could be explained more clearly. The bootstrap elements need fleshing out, and the weighting procedures could be explained with greater clarity.

In the last paragraph before quartet analysis (page 14), the null hypothesis for the bootstrap method (a different bootstrap method to the one used previously?) is that there is less morphological convergence than in molecular data. Should this rather be that they are equal?

In their analysis and discussion on bats, there is no indication of how many of each kind of character are removed when they prune out the convergence prone ones.

In general, the paper would benefit from a careful editing, with particular attention to English usage and tenses. I have made a few nods in this direction, but haven't attempted to do this throughout the manuscript.

The advent of molecular biology supplied (not supplies).

Muddled sentence: 'Second, phylogenetic analysis including fossils helps understand evolutionary relation, time, and process of fossils as well as extant species'.

Introduction: I understand the shorthand of using 'morphology' to refer to all non-molecular characters, but it could be a bit misleading. Perhaps just 'non-molecular'?

I'm not sure about references to the "true" tree for the empirical data.

Response to the reviewers

We would like to thank the two reviewers for their critical yet constructive comments, which helped improve the rigor and clarity of our manuscript considerably. Below we address these comments.

Reviewer 1:

Comment 1

This is a well written manuscript that outlines a new approach for improving the accuracy of phylogenetic analyses with morphological data. While the general approach is interesting, the analyses are insufficient to justify the main conclusions. I have several comments/suggestions for improvement. Some are major and other are minor.

Response:

Thanks to the reviewer for the overall positive evaluation.

Comment 2

In the first sentence of the Discussion the authors state the following: "Our analysis of comparably large numbers of morphological and molecular characters previously used in phylogenetic inference showed that morphological characters experienced more convergent evolution than molecular characters, confirming a long-held belief of the phylogenetics community." This conclusion is not general and is based on the authors' comparison of morphological data and amino acid characters from O'Leary et al.'s (2013) mammalian data set. However, not all types of molecular data are the same and it is not clear that this conclusion holds for different types of molecular data. The authors should perform a comparison for morphological and DNA data from O'Leary et al. to determine if the same pattern holds. Unlike amino acids, which have 20 possible character states, there are only four possible character states (nucleotides) for DNA and one may expect to find higher levels of convergence than for amino acids. Also, the majority of substitutions at the DNA level for protein-coding genes are synonymous (silent) substitutions that do not result in a different amino acid. Together, a smaller number of possible character states (4) plus silent substitutions may result in higher levels of convergence for DNA characters than for morphological characters. This should especially be the case for third positions of DNA codons since these will have the highest levels of silent substitutions. The authors should perform a separate analysis with 3rd codon positions. If the tree based on 3rd positions remains closer to the full molecular tree (based on 1st, 2nd, and 3rd positions) but has similar or higher levels of convergence than morphological characters then the authors will need to re-evaluate their conclusions.

Response

We followed the suggestion to compare morphological and DNA sequence data from O'Leary et al. with the same methods used to compare morphology and amino acid sequences. The nucleotide sequence-based tree in O'Leary et al. was used as molecular tree and 19,227 parsimony informative nucleotide sites were used as molecular characters. In whole-tree analysis, morphological characters have higher convergences per character and higher Cv/Dv ratios than nucleotide sites, regardless of the tree topology used. In quartet analysis, morphological characters again have higher convergences per character and higher Cv/Cs ratios than nucleotide sites. Furthermore, we confirmed that both Cv/Dv and Cv/Cs of nucleotide characters are negatively correlated with the number of states. Hence, the new results are completely consistent with our previous results. Notably, the median number of states for nucleotide sites is still significantly higher than that of morphology (3 vs. 2, P -value $< 10^{-300}$), which explains why convergence is lower in nucleotide sites than morphologies. We have added the new analysis (pages 6-8) and results (Figs. S4, S5c) to the manuscript.

Following the suggestion, we built a parsimony tree of the 46 extant species using only third codon positions. This tree has a drastically different topology ($d_{RF} = 1.0$) from both the morphological tree and molecular tree, and has low level of bootstrap support ($< 70\%$) for many internal nodes. Consistently, we found the convergence level of 3rd codon positions much higher than the 1st and 2nd codon positions. Thus, these findings are consistent with our conclusions. Nevertheless, due to the space limit and the fact that almost no one uses third codon positions alone in phylogenetics, we decided not to include the above results in the manuscript.

We agree that our conclusions should be further scrutinized with other large datasets that may become available in the future, and have added this caveat to Discussion (page 11, paragraph 2).

Comment 3

After proposing a new method to filter out morphological characters with high levels of homoplasy, the authors apply this method to O'Leary et al.'s morphological data set and conclude that it is useful for improving the performance of phylogenetic analyses with morphological characters. However, they only show the tree for bats. The authors should show the full tree based on their "curated" morphological data set. Does their method eliminate major problems with O'Leary et al.'s morphological tree that were discussed by Springer et al. (2013, reference 1 in Zou and Zhang manuscript)? These problems include numerous polyphyletic groups such as an "insectivore" group, a "spiny hedgehog" group, an "ant & termite eating" group, a "tree dwelling group", "an ungulate group", and a "cud-chewing" group. Further, does the "curated" morphological data set result in the recover of natural clades such as Afrotheria, Boreoeutheria, Euarchontoglires, and Laurasiatheria? It is important to show the full results so that the reader can determine if the approach proposed by the authors is generally effective for improving cladistic analyses with morphological characters. I would further suggest that the authors apply

their method to another very large morphological data set for mammals to test its general effectiveness for improving the accuracy of cladistic analyses with morphological data. Specifically, the discrete morphological data set of Halliday et al. (2015, *Biol. Rev.* doi: 10.1111/brv.12242). Halliday et al.'s tree based on morphology alone has numerous problems (e.g., polyphyly of Afrotheria, Laurasiatheria, and Euarchontoglires; talpid diphyly; elephant shrew diphyly, etc.) and it would be useful to know if the authors' new approach is useful for fixing these problems.

Response

Following the suggestion, we now show the full tree topology of the combined tree before and after removing high-convergence characters in Fig. S8. Instead of comparing with the morphological tree, we think the improved tree (Fig. S8b) should rather be compared with the original total evidence tree (Fig. S8a), which is the tree preferred by O'Leary et al. and the baseline before improvement. The improved tree has two notable differences from the original total evidence tree: (i) two origins of echolocating bats, and (ii) grouping of pangolin with carnivores. Because we do not assume that the molecular tree is correct, we discuss these topological changes as potential improvements. Nevertheless, the improved tree largely agrees with the molecular tree in order- and superorder-level groupings mentioned by the reviewer. It should be noted that, as shown by our computer simulation, our method improves the accuracy of tree inference but cannot remove all phylogenetic errors. We have added these discussions to page 14.

Our method is not applicable to the Halliday et al. dataset, because our approach requires comparing molecular sequence data with morphological data to identify consistent quartets, but Halliday et al.'s dataset has only morphological characters. We did add a note that our results should be further tested in the future with more data and more taxa (page 11, paragraph 2).

Comment 4

Page, 7. "Because the vast majority of molecular convergences are explainable by chance, the smaller Cv/Dv and Cv/Cs ratios of average morphological characters when compared with molecular characters of the same numbers of states suggests that most morphological convergences observed in the data analyzed are probably also attributable to chance." The logic behind this statement is not transparent. Instead, the major problem with morphological data sets such as O'Leary et al. may have more to do with the non-random pattern of homoplasy rather than the amount of homoplasy. Specifically, is homoplasy randomly distributed across the tree or are there correlated patterns of homoplasy that drive the overall results? The recovery of polyphyletic groups such as the "ant & termite eating" group (and others, see above) suggests that correlated homoplasy (i.e., convergence not by chance) is perhaps the major problem with morphological data. For example, ant and termite eating forms tend

to have reduced or absent teeth, long snouts, long tongues, forelimbs adapted for digging, etc. It is not clear how Cv/Dv or Cv/Cs indices can be used to differentiate between the amount of homoplasy and the pattern (random versus non-random) of homoplasy.

Response

Here is the logic. Convergence may happen by chance or by adaptation. In amino acid sequences, it has been shown that the amount of convergence observed is fully explainable by chance. Since morphological characters have no more convergence than amino acid sites after the control of the number of states, morphological convergence is also explainable by chance. Of course, this does not mean that there is no adaptive morphological convergence, but does imply that most morphological convergences in the dataset analyzed are probably by chance.

The reviewer seems to believe that phylogenetic errors can be caused only by adaptive convergence (i.e., correlated convergence) but not by random convergence. To test this idea, we compared the morphological and molecular trees we built in the simulation section of the manuscript to the true tree used in the simulation. We found that the morphological tree has a significantly larger d_{RF} than the molecular tree. Because there is no correlated convergence in the simulation, our result indicates that random convergence can lead to wrong morphological trees. When morphological data generate a wrong tree, there must be some morphological traits that support the wrong grouping. So, the observation of the ant and termite eating group and other such groups does not by itself indicate correlated convergence. Nevertheless, the ant and termite-eating related traits mentioned by the reviewer are apparently adaptive and correlated. Interestingly, these traits do not exist in the morphological dataset of O'Leary et al. Many traits that are known to be subject to adaptive convergence had been removed by O'Leary et al. because they apparently interfere with phylogenetic inference. So, it is important to note that our conclusion about random convergence applies to O'Leary et al.'s dataset, which represents a morphological dataset for phylogenetics, but may not apply to all morphological traits. We examined the morphological traits that group the four ant- and termite-eating species in the dataset. We found that these traits are unrelated to ant- and termite-eating and are not specific to these four species.

We have added the above points to the manuscript (pages 12-13).

Comment 5

My understanding of the simulations performed by the authors is that they were based on random homoplasy and ignored potential problems with correlated homoplasy as may occur for real morphological data.

Response

Yes. Our approach for improving phylogenetic accuracy will not work if most convergences are nonrandom but are concentrated in a small group of taxa (correlated convergence). Nevertheless, our results suggest that most convergence is random, as described above. We clarified this in Discussion (page 14, paragraph 1).

Comment 6

The authors should comment on their results in the context of Sansom and Wills' 2013 paper (Sci. Reports 3, 2545) on stem-ward slippage, where fundamental taphonomic biases for skeletal characters can cause fossils to be recovered as erroneously primitive.

Response

We noticed two facts in Sansom and Wills' paper that are worth commenting on. First, in the meta-analysis, the authors do not know the true tree topology, so it is not clear whether the shift is due to biases in soft tissue characters or fossil characters. Second, the upward shift of fossilized taxa is statistically significant but with small effect size; many fossilized taxa actually shifted towards the leaves. Given these facts, the relevance of the findings to our paper is weak. Nonetheless, we cited this paper (page 15) because it is possible that traits that can be studied in fossils are biased.

Reviewer 2:

Comment 1

This is a good paper containing important results that will be of great and widespread interest. I hope that it will find a home in Nature Communications. There are several issues that I think the authors should consider, however.

Response

We thank the reviewer for the overall positive evaluation.

Comment 2

A. Summary of the key results

The authors demonstrate that their sample of morphological characters contain more homoplasy on average than molecular ones. They also show that this is a function of a more limited character space in the case of the former. This last is a surprising result, as it has often been assumed that signal saturation and multiple hits are problems more particularly bedeviling nucleotide data than morphological character states (where the possibilities for character state innovation are usually believed to be more open-ended). In that context, I'm not sure about the statement in the abstract that convergence "...is widely believed to be rarer for molecular sequences than for morphologies."

Response

We changed the expression “is widely believed” to “is believed” and cited publications with this belief. We mentioned in Discussion (page 13) that morphological characters used in phylogenetics may have fewer states than average morphological characters because characters with many states may be hard to curate and thus are avoided in phylogenetics, and provided a reference. Our newly added analysis of nucleotide sequences further supports our conclusion (see response to comment 2 of reviewer 1).

Comment 3

B. Originality and interest: if not novel, please give references

This is an original contribution, and will be of great interest to all evolutionary biologists and systematists in particular. It has often been assumed that molecular characters have more desirable properties than morphological characters for phylogenetic inference, not least their abundance, some elements of superior objectivity and increasingly sophisticated frameworks for their analysis. This paper demonstrates that this does indeed seem to be the case, and represents an important result entirely appropriate for publication in Nature Communications.

Response

Thanks for these positive evaluations.

Comment 4

C. Data & methodology: validity of approach, quality of data, quality of presentation

The main empirical conclusions of the paper are derived from a single, albeit very large, data set for mammals. The simulations bolster these conclusions, but the authors may wish to consider whether there might be benefits from analyzing a wider sample of empirical cases. I am not suggesting that they should do this, but I think there needs to be some discussion of this limitation in the paper.

Response

Indeed we analyzed only one very large dataset. Because results from different analyses of this dataset are consistent with one another, our findings are likely to be robust. Nevertheless, we agree that our conclusion should be further scrutinized in the future by more data. We added this caveat to Discussion (page 11).

Comment 5

D. Appropriate use of statistics and treatment of uncertainties.

Subject to C above, the stats and methods seem sound.

Response

See response to comment 4.

Comment 6

E. Conclusions: robustness, validity, reliability

Again, one might wish to add some circumspect qualification that this is just a single empirical data set. It is entirely possible - even probable - that the pattern they observe is a general one. I wonder whether their conclusions (and the title of the paper) might be better couched in terms of mammalian phylogeny, for which they have the largest and most impressive data set available. They could then go on to set out the case that this likely to be a generality. Investigating this possibility really is a hugely pressing research agenda, and this paper makes a fantastic, novel and excellent contribution to it.

Response

Thanks for the positive evaluation. We hope that large datasets like the one analyzed here will become available in other species so that similar analyses can be conducted in more species. After careful consideration, we decide to keep the title unchanged, because we believe that the methodology developed here is general, although we analyzed only one very large dataset of mammals. We did add "mammals" to Abstract to make our results and conclusion more specific.

Comment 7

F. Suggested improvements: experiments, data for possible revision

Analysing more data sets is probably impractical, and represents a lot of additional work that I think it would be unfair to ask the authors to do.

Perhaps there needs to be some exploration of how their approach for removing convergence-prone characters compares with methods that remove or otherwise downweight characters prone to homoplasy more generally. Implied weighting, successive approximations weighting and even clique analysis have this effect with varying degrees of severity, in addition to the normal practice of omitting morphological characters a priori that are believed to largely convey noise at the level of the analysis. How do these approaches compare? I am not necessarily suggesting any additional analyses here. Homoplasy comprises convergence plus reversal, and there may be a nuanced argument that the former is intrinsically less likely than the other: hence the advantage of removing convergence-prone characters rather than those subject to high levels of homoplasy more generally. 'Dollo' parsimony, for example, forces all homoplasy to take the form of reversals, and requires a rooted tree rather than a network for its computation.

Response

Weighting methods generally have the problem that weights are determined more or less subjectively. For example, successive approximations weighting relies on the original tree that is based on all characters to determine the reliability and thus the weight of each character. Implied weighting requires an artificially assigned parameter k . Removing convergence-prone characters is equivalent to setting the weights of these characters to zero. In the future, it may be worth studying whether giving different weights to characters with different levels of convergence further improves phylogenetics.

We did not explicitly study reversal in this paper. Nevertheless, high-convergence characters are also expected to have high reversals if convergence is mostly by chance owing to the availability of few states. Thus, our method may have alleviated the problem of reversal as well.

These points have been added to Discussion (page 14).

Comment 8

G. References: appropriate credit to previous work?

Yes, all fine here.

Response

Thanks.

Comment 9

H. Clarity and context: lucidity of abstract/summary, appropriateness of abstract, introduction and conclusions

Good. I had to read the paper right through before I understood the approach and what had been done properly (with some issues outstanding) and it may be possible to make this clearer earlier in the manuscript.

Response

We have made results more accessible and have added references to Methods when appropriate.

Comment 10

In the introduction, the authors could state more clearly whether or not they are in favour of using morphological characters. Convergence isn't just a problem for morphological data. It's one of the main issues bedeviling all phylogenetic analyses. The authors also make the point that most morphological and molecular trees are concordant, suggesting the situation is less black and white than some of their other statements might suggest.

Response

We think that morphological characters can be useful in certain situations, as we wrote in Introduction “Second, phylogenetic analysis that includes fossils can help understand evolutionary relation, time, and process for fossils as well as extant species. Because molecular characters are inaccessible in the vast majority of fossils, knowing the frequency of morphological convergence is critical to assessing the reliability of phylogenies involving fossils.” Actually, this study itself is partly aimed at evaluating whether morphological characters are useful in phylogenetics. We changed “morphological and molecular trees are largely concordant with each other” to “although morphological and molecular trees are often concordant with each other, this is not always the case”. We believe this altered sentence better reflects the reality.

Comment 11

The method for assessing the amount of convergences on branches could be explained more clearly. The bootstrap elements need fleshing out, and the weighting procedures could be explained with greater clarity.

In the last paragraph before quartet analysis (page 14), the null hypothesis for the bootstrap method (a different bootstrap method to the one used previously?) is that there is less morphological convergence than in molecular data. Should this rather be that they are equal?

Response

We have clarified the amount of convergences on branches, the bootstrap, and the weighting procedures in Methods.

For bootstrapping, we want to confirm that the excess of morphological convergence relative to molecular convergence is not simply by chance. This is a one-tailed test, so “less than” is used. Use of a two-tailed test does not alter the conclusion.

Comment 12

In their analysis and discussion on bats, there is no indication of how many of each kind of character are removed when they prune out the convergence prone ones.

Response

We add the numbers of removed morphological and molecular characters as suggested. 1007 morphological characters and 2923 molecular characters remained. Hence, 2407 and 2799 were removed, respectively.

Comments 13

The advent of molecular biology supplied (not supplies).

Muddled sentence: 'Second, phylogenetic analysis including fossils helps understand evolutionary relation, time, and process of fossils as well as extant species'.

Introduction: I understand the shorthand of using 'morphology' to refer to all non-molecular characters, but it could be a bit misleading. Perhaps just 'non-molecular'?

I'm not sure about references to the "true" tree for the empirical data.

Response

We thank the reviewer for these suggestions, and we have made the modifications accordingly.

For the definition of “morphology”, we stated in the beginning of Introduction that “morphological, physiological, or behavioral characters, collectively called morphological characters hereinafter...” Because most of the non-molecular characters used in this study are morphological, we believe that using "morphological" is more specific than "non-molecular".

For the true tree of empirical data, we were referring to the unknown true species tree of the species under study.

Reviewers' Comments:

Reviewer #2 (Remarks to the Author)

The authors have done a good job of taking most of my comments on board, and appear to have responded well to those of the other referees.

I hope to see this excellent contribution published soon.

Two issues still stand out, however.

Comment 6

The work is still based on a single (albeit very large) data set for mammals. I feel strongly that the title should reflect the nature of the study, and should therefore make reference to mammals.

It may well transpire that their conclusions are general ones across all major groups and at all levels of taxonomic resolution. Indeed, I think this likely, but to claim so at this stage (even qualified in the abstract) is premature.

This is simply rebranding.

Comment 4

"Because results from different analyses of this dataset are consistent with one another, our findings are likely to be robust."

The plurality of methods is excellent, but these methods are all intended to address the same question. The issue of the robustness of the conclusions for this data set (which they nicely lay to rest) is different from the issue of the generality of the conclusions across groups, taxonomic levels and timescales - which is the point of the comment. This is solved as above.

Response to Reviewer

Reviewer 2

Comment 1:

The authors have done a good job of taking most of my comments on board, and appear to have responded well to those of the other referees. I hope to see this excellent contribution published soon.

Response:

Thanks!

Comment 2:

Two issues still stand out, however.

The work is still based on a single (albeit very large) data set for mammals. I feel strongly that the title should reflect the nature of the study, and should therefore make reference to mammals. It may well transpire that their conclusions are general ones across all major groups and at all levels of taxonomic resolution. Indeed, I think this likely, but to claim so at this stage (even qualified in the abstract) is premature. This is simply rebadging.

Response:

Per the reviewer's suggestion, we changed the title to "Morphological and molecular convergences in mammalian phylogenetics".

Comment 3:

"Because results from different analyses of this dataset are consistent with one another, our findings are likely to be robust."

The plurality of methods is excellent, but these methods are all intended to address the same question. The issue of the robustness of the conclusions for this data set (which they nicely lay to rest) is different from the issue of the generality of the conclusions across groups, taxonomic levels and timescales - which is the point of the comment. This is solved as above.

Response:

We agree, and have changed the title to address this comment. See response to Comment 2. We have also changed the titles of Fig. 1 and Fig. 2 to emphasize this point.